# When the Tension Is Rising: A Simulation-Based Study on the Effects of Safety Incentive Programs and Behavior-Based Safety Management

**Sebastian Brandhorst *** and **Annette Kluge**

Faculty of Psychology, Ruhr-Universität Bochum, 44801 Bochum, Germany; annette.kluge@rub.de
* Correspondence: sebastian.brandhorst@rub.de; Tel.: +49-176-6373-9567

**Abstract:** When an organization's management creates a goal conflict between workplace safety and the profitability of the organization, workers perceive work-safety tension. This leads to reduced safety-related behavior, culminating in higher rates of occupational injuries. In this study, we explored design components of behavior-based safety programs: audit results and process communication, reward and punishment, and the framing of production goals as gains or losses. This allowed us to directly observe the effects of the goal conflicts and of the countermeasures that we designed in this study. We examined the perceived work-safety tension using a simulated water treatment plant in a laboratory study with 166 engineering students. Participants had the task of conducting a start-up procedure. The operators' goal conflict was created by a choice between a safe and mandatory (less productive) procedure and an unsafe and forbidden (more productive) one. As participants were told that their payment for the study would depend on their performance, we expected that rule violations would occur. We found acceptance of measures and their design as important for rule related behavior. Work-safety tension emerged as a strong driver for violating safety rules. We conclude that safety incentive programs can become ineffective if goal conflicts create work-safety tension.

**Keywords:** safety-related rule violation; occupational safety; goal conflict; audit consequences; audit feedback; work-safety tension

## 1. Introduction

Under the influence of economic pressure in the chemical production industry, safety-related rules are frequently violated, which can result in serious incidents, such as the emission of several tons of corrosive chemicals in Bhopal in 1984, killing an estimated 20,000 people and injuring about half a million [1]. The consequential damage to people and the environment is almost impossible to assess. A further example is the explosion of the BP refinery in Texas City in 2005 [2]. The tragedy cost the lives of 15 workers and had similar economic causes to those in Bhopal. Approximately 30% of the causes of accidents in Texas City can be attributed to a failure of safety management [3]. There are several methods to analyze the root causes of accidents to support and improve safety management [4]. However, there are different perspectives on how the findings of accident analysis should be implemented, regardless of the systematic approach used to obtain the findings.

In the last few years, incentive programs, which are an aspect of behavior-based safety management (BBSM), have garnered increased public awareness. Unions criticize BBSM programs as "blame the worker" programs and predict that "This BS will kill you"(author's note: BS = behavioral safety) [5]. In addition, some studies indicate that near-accidents are not sufficiently reported because of the prevailing (inadequate) safety and reporting culture [6–8]. In 2012, the US Government Accountability Office (GOA)

called for better guidelines with regard to safety incentive programs. In particular, rate-based programs, which reward workers who have the fewest reported accidents within a given time period, are suspected to discourage the reporting of safety-related incidents [9]. At about the same time, the Organizational Health and Safety Administration (OSHA) published a whistleblower memorandum highlighting problematic employers' policies that discourage workers from reporting illnesses or injuries [10]. The GOA report [9] contained a comment from the OSHA that announced upcoming guidelines for safety incentive programs. Implemented in the updated version of 2016 [11], these guidelines were heavily criticized by employers' representatives [12] for restricting the options for incentive programs and for impeding employers' ability to carry out drug testing following reports of injuries. Furthermore, unions refer to lawsuit verdicts that assert that an implemented behavior safety program blames workers for their accidents instead of preventing them from happening in the first place [5,13]. In 2018, OSHA published an interpretation letter on its 2016 guidelines [14], which pointed out that safety incentive programs can be beneficial to safety as long as they do not discourage workers from reporting accidents. Occupational accidents are the result of a combination of active failures (slips, lapses, mistakes) and latent failures (understaffing, maintenance failure [15]), underlining employers' responsibility to ensure that active failures will not result in an accident.

In summary, the debate of stakeholders regarding BBSM programs suggests that interventions, which lead to intended safety behavior, might come with the cost of unintended side effects. What do empirical results suggest?

In our study, we aim to contribute to the discussion with empirical data and investigate the influence of BBSM interventions on workers' perceived work-safety tension and their compliance with safety-related rules. We explore feedback mechanisms in a simulation-based experiment after safety audits and the effect of financial punishment and reward for safety-related behavior. Overall, BBSM measures are intended to reduce the perceived work-safety tension. With the examined measures (feedback-based interventions), we can determine their internal validity and, on this basis, apply the results to the real context. The measures aim to reduce this tension in order to strengthen safety-related behavior. We aim to provide insights into the effectiveness of the explored measures and the best approach to designing a work environment that reduces work-safety tension.

## 1.1. Safety-Related Rule Violations and Organizational Behavior Management

BBSM is part of the broader concept of Organizational Behavior Management (OBM), which has its origins in the 1960s [16] and arose from the behavioral approach of Thorndike [17], Taylor [18], and Skinner [19].

The scope of OBM covers performance management, system analysis, and behavior-based safety [16]. Our research is focused on the third aspect. The behavioral approach is realized in the design of the environment, which leads to the safe behavior of workers through increased safety awareness. Behavior-based interventions that aim to elicit a desired behavioral change can be classified into antecedent-based (e.g., task clarification) and consequence-based (e.g., performance feedback) interventions. Based on research from the last 30 years, Ludwig [20] concluded that the most effective way to achieve a change in safety-related behavior is through a combination of both types of intervention. The broad spectrum of behavioral change techniques is both a great strength and a major weakness of OBM. Based on the seven feedback dimensions described by Alvero, Bucklin, and Austin [21], the feedback tool alone results in 338,688 possible pairings if all characteristics of the seven dimensions are combined. This enables a great adaptability to specific demands, but in terms of analyzing the effects of feedback interventions, it is difficult to compare and generalize the results.

In this study, we focus on two aspects of feedback communication: (1) results versus process feedback and (2) production outcomes displayed as a loss or gain in relation to the intended production goals. In previous studies, we focused on the communicated probability of receiving an audit [22] and the timing of an audit [23]. The results showed

that when participants received precise information about audit probabilities, they tended to show a lower safety performance. Likewise, safety performance was lower when audits occurred a long time after they had been announced, and it was higher when audits took place directly following their announcement. Displaying the production outcome to the worker constitutes permanent feedback, which is not listed in Alvero et al.'s dimension of feedback frequency [21]. In previous studies, we displayed the production feedback in terms of gain and loss framing [23] and found lower safety performance in the loss-framing condition, but there was no detected interaction with the manipulation of audit timing or the communicated audit probability.

With respect to Alvero et al.'s [21] dimensions, result and process feedback can be described in written and graph form (medium), on a monthly basis (frequency), individually (participant), privately (privacy), or as a comparison of an individual's performance with a standard of individual performance (content, only applicable for process feedback), and it can be automated (source, not covered by that dimension). In this study, we also integrated the framing in the display of the production outcome while manipulating the content of the audit.

### 1.2. Behavior-Based Safety Incentive Programs

The behavior-based safety management approach was further developed into the behavior-based safety incentive program (BBSIP [24]). However, it should be emphasized that behavioral safety is not limited to incentives or mere compliance with rules. The BBSIP represents a section of the spectrum of behavioral safety approaches. Within the BBSIP, employees are rewarded for safety-related behavior through financial or other incentives, in contrast to rate-based safety incentive programs, in which employees are rewarded for achieving organizational goals (e.g., number of days without an accident [24]). As described in the introduction, current public and scientific discussion revolves around behavior- and rate-based incentives, which can be distinguished by the role of organizational goals: a rate-based approach may be dysfunctional if the organizational goal is itself contradictory to safety goals. Being productive and working under time pressure to achieve organizational goals conflict with safety goals and lead to unsafe behavior [25]. However, if the management is committed to safety goals (over production goals), safety incentives become effective [26,27]. The type of incentive can vary from monetary to a broad range of non-monetary forms, such as recognition, time off, special assignments, advancement, increased autonomy, training and education, or social gatherings. To generate a positive effect, the incentive should fit with employees' personal values [28]. In summary, the effectiveness of BBSM relies on various levels, beginning with the type and goal of the intervention, depending on the safety culture and organizational commitment, and determined by personal values when it comes to its acceptance.

In this study, we focus on goal conflicts (productivity vs. safety) and the organization's position on safety at work, as compliance with safety regulations entails personal disadvantages for the participants. This scenario depicts actual work environments and has led to, among other issues, the controversy surrounding rate-based incentive programs.

### 1.3. Hypotheses

Although two meta-analyses [21,29] have examined the type of feedback that is the most effective, this question still remains unanswered, largely due to the broad range of operationalizations within the studies on which the analyses were based. In general, it seems that feedback is most effective if it is combined with monetary and non-monetary consequences and goal setting [29] and applied in the context of training, if task clarification is provided in graphical and written form, and if feedback is provided on a daily and weekly basis [21]. With regard to the mode of feedback for behavior-based safety management, which is the audit result in the present study, we aim to answer the following question: Should the feedback relate to (a) compliance with a rule (yes/no) or (b) optimizing the application of the rule? In a study by Komaki, Barwick, and Scott [30], participants were

told which behavior was concretely expected in an intervention, and Pichler, Beenen, and Wood [31] showed in a meta-analysis that feedback appraisal reactions are influenced by the knowledge of performance standards. Optimizing the application of the rule appears to be particularly relevant, given that our previous studies [32] and the studies by Christian et al. [33] and Griffin and Neal [34] revealed that low performance and little knowledge lead to more safety-related rule violations. The question, therefore, arises as to whether it is possible to reduce safety-related rule violations by presenting process-related feedback. Our first hypothesis thus proposes the following:

**Hypothesis 1 (H1).** *Process-related feedback will lead to more safety-rule compliance than feedback that only presents results.*

Current research shows that in the context of work performance, monetary reinforcement is more accepted than punishment [35–39]. In our previous research, participants were financially punished (loss of acquired salary) for rule-deviant behavior (applying an unsafe procedure) if this behavior was detected by audits [22,23]. Accordingly, a question arises surrounding the effects of rewards when rule-compliant behavior is reinforced. Based on this question, we hypothesize the following:

**Hypothesis 2 (H2).** *Rewards for safety-related rule compliance will lead to more rule compliance than punishment of safety-related rule violation.*

With regard to environmental influences on rule-related behavior, we investigated the framing of production outcomes. The prospect theory of Kahnemann and Tversky [40] postulates that risk aversion occurs when it comes to gains, and risk-seeking behavior arises in the face of losses [41]. In the majority of previous studies, we found empirical support for risk-averse and rule-compliant behavior in the context of the gain framing of production and for risk-seeking behavior in loss framing conditions [42], and other results support the predictions of the prospect theory [22,23,43,44]. In line with the majority of our previous findings regarding the framing effect, we assume the following in our third hypothesis:

**Hypothesis 3 (H3).** *Participants in a loss-framing condition will violate the rule more often than participants in a gain-framing condition.*

Besides the design of safety programs, we evaluated the perception of work-safety tension. This construct describes a perceived discrepancy between safety and production goals [45]. While there is no general consensus among safety researchers regarding the operationalization of a safety climate [46], work-safety tension as a facet of the safety climate is the most suitable for predicting unsafe behavior [47]. However, in contrast to the large amount of effort that has been undertaken to research and operationalize safety climates, the promising facet of work-safety tension as a valuable predictor of safety-related behavior has received less research attention. To replicate the findings of a field study [46] in an experimental setting, our fourth hypothesis is as follows:

**Hypothesis 4 (H4).** *The higher the perceived work-safety tension, the greater the number of violations that occur.*

## 2. Materials and Methods

To determine the sample size, we used G*Power 3.1 under the assumption of f = 0.25 for 8 groups and calculated a total sample size of N = 360 (*n* = 45 subjects per group). Based on experiences from previous studies, in which effects became visible at a sample size of *n* = 20, and due to financial considerations given that the study lasted for 4.5 h, we aimed to include 20 subjects in each group. All participants (N = 166, age *M* = 22.70 years, *SD* = 3.17, 54 female) were recruited at the University Alliance Ruhr, Germany, which comprises

the campuses of the Universities of Dortmund, Bochum, Essen, and Duisburg. To ensure external validity by selecting students who might potentially go on to work as control room or field operators in the process industry, all participants were students of engineering sciences. The participants were told that the study's purpose was to evaluate a training intervention for industrial wastewater treatment plants. Moreover, they were told that their payment for participation would depend on their productivity, such that they could earn up to EUR 50 in the 4.5 h study. In fact, all participants received EUR 50 at the end of the study. Participants signed an informed consent form, which was approved by the ethics committee (No. 189) of the Faculty of Psychology, Ruhr-University Bochum.

### 2.1. Applied Simulation and Experimental Environment

The experimental setting mirrors the job of a control room operator in a company called WaterTec. All participants were trained to be a control room operator for one of 20 plants of the (fictional) company WaterTec. The participants' task was to start up the plant, which separates a solvent–water mixture into its components. The simulation is called Waste Water treatment Simulation (Version 3, RUB, Bochum, Germany) (WaTrSim [44], Figure 1) and has been further developed for the purpose of investigating rule-related behavior since 2013. In order to manipulate conditions that have an impact on safety-related behavior, it is essential to use simulations due to practical and ethical considerations. The software simulates a whole production year, encompassing 10 training weeks and 48 production weeks (58 simulated weeks in total), with the production weeks divided into 4 quarters of 12 weeks each. Every simulated week lasts for 2 min. The participants' task was to start up the plant, which consists of 10 steps and requires action and monitoring.

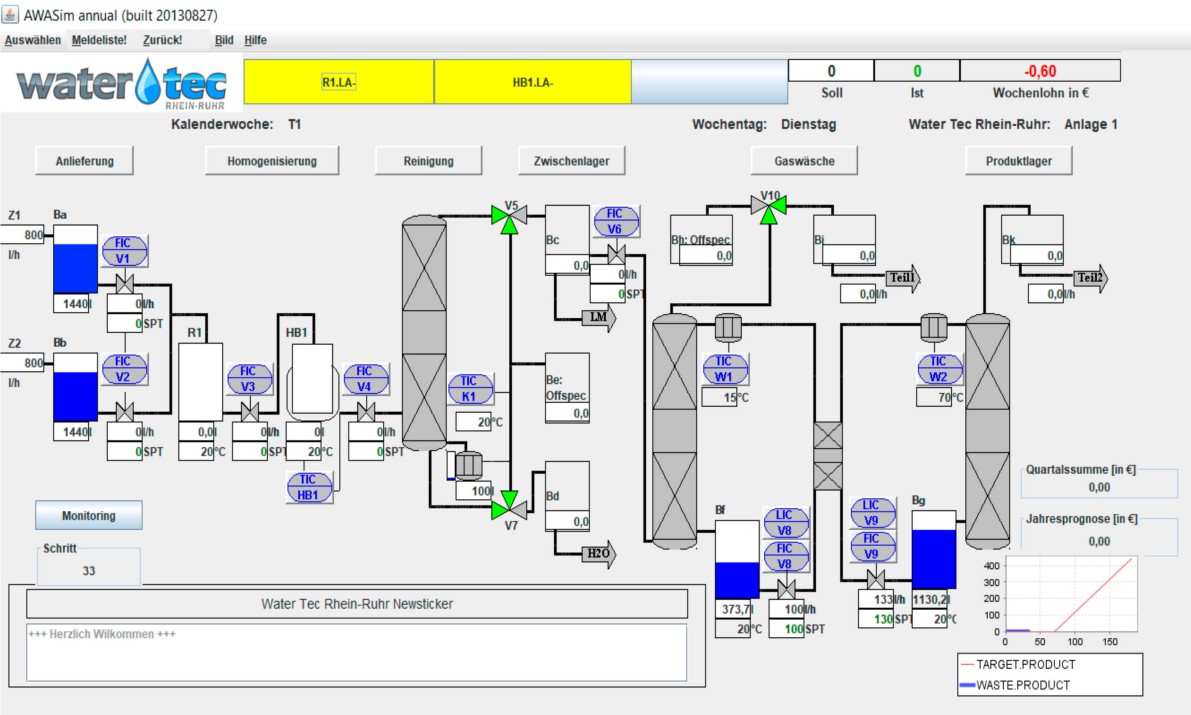

**Figure 1.** Interface with valves, heaters, tanks and salary displays.

The performance of start-up procedures for process control tasks can be classified as normal and non-routine but infrequent and can therefore be seen as rule-based behavior [48]. Conflicting rules and goal conflicts are the strongest driving forces of non-compliant behavior [49].

*2.2. Operationalization of Goal Conflict and Rule Violation*

To induce a goal conflict, participants were told that their payment would depend on their productivity. Before the first simulated quarter, the participants were trained to start up the plant using a productive (but unsafe) procedure, which would enable them to generate a weekly salary of EUR 1.00. This was the maximum possible salary to be earned in a week. After the first simulated quarter, the experimental intervention took place. Participants were informed that an accident had occurred in one of the company's plants, that the safety department had, therefore, extended the start-up procedure to prevent future accidents, and that this new extended procedure had now been declared as a mandatory safety rule (safe procedure). The safe procedure consisted of three additional steps, meaning that the start-up took more time and led to lower production within the available time. It also meant that by being safe and behaving in compliance with the new rule, participants could only earn a maximum of EUR 0.80. This goal conflict between productivity and safety thus directly affected the participants' salary.

The additional steps of the safe procedure were communicated as being (and indeed are) safety-relevant. If the participants decided to apply the productive but unsafe procedure to receive a higher payment, the simulation software was able to detect the missing steps. This software-driven monitoring of the participants' behavior is important for counting the number of violations of each participant, but it is also relevant to our audit-based hypotheses.

The announcement of the safe procedure was accompanied by a note that several audits would be conducted at random. Previous findings show that precise information about the frequency of audits evokes the illusion of control, resulting in rule violations [22]; thus, no precise information on the frequency of audits was provided in the present study. In fact, nine audits were implemented, and they were conducted in the exact same production weeks for all participants in order to eliminate the effect of the sequence as a potential explanation for any effect. The audits were implemented and communicated as checks of the participants' rule compliance. The audit result on rule violation or rule compliance was automatically generated according to the participants' actual behavior. The audit modes are described in detail in the paragraph below.

*2.3. Independent Variables*

The study employed a 2 × 2 × 2 between-subjects design with feedback communication (result versus process, Hypothesis 1), audit consequences (reward versus punishment, Hypothesis 2) and framing (loss versus gain, Hypothesis 3) as the factors.

2.3.1. Feedback Communication: Result versus Process Feedback

Besides the different results (positive and negative feedback) and the consequences (punishment and reward), the mode of feedback also differed (Figure 2). The result feedback referred only to whether participants violated or complied with the rule.

The process feedback comprised an extended pop-up window that showed the chosen procedure (safe or unsafe procedure) presented as a list. Each step was highlighted in green, orange, or red, which distinguish three characteristic combinations that refer to the order and applied values.

- A step highlighted in green was conducted in the correct order with the correct value.
- A step highlighted in orange was conducted either in the wrong order or with the wrong value.
- A step highlighted in red was conducted both in the wrong order and with the wrong value.

The window also showed four texts, which differed depending on the chosen procedure and whether the chosen procedure was correctly conducted (unsafe procedure correct/incorrect and safe procedure correct/incorrect).

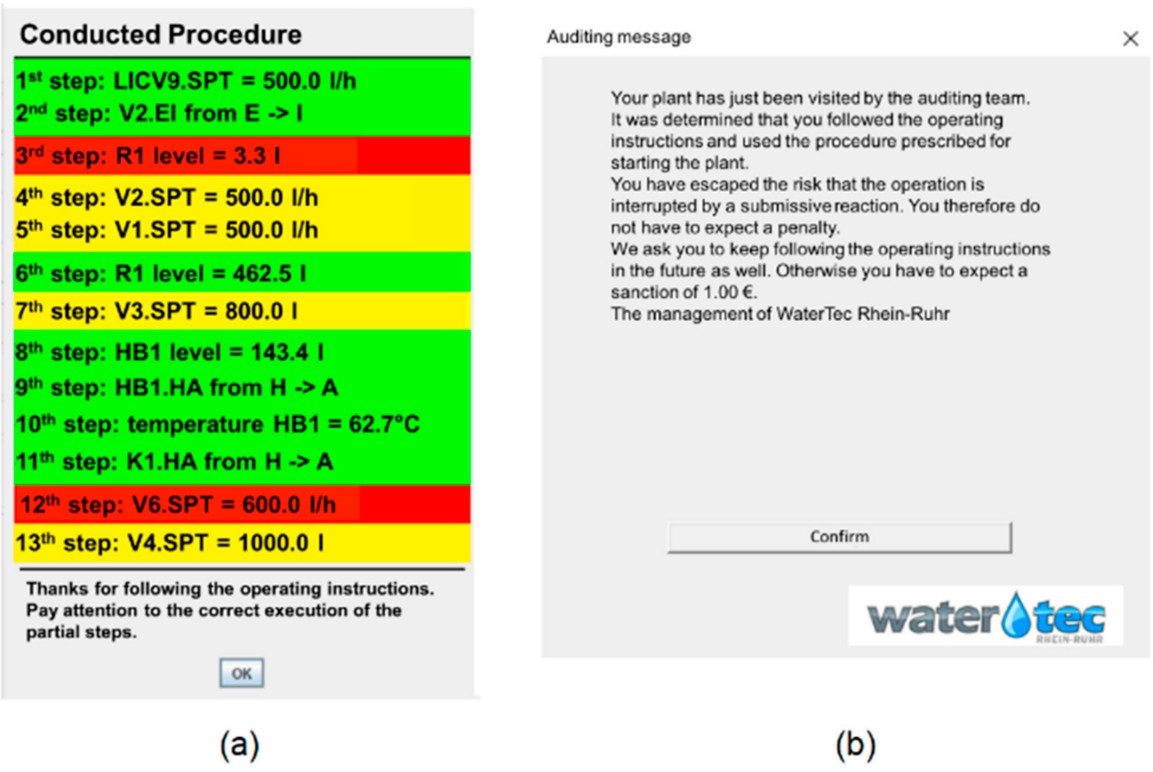

**Figure 2.** Process feedback (**a**) showing the actual conducted procedure rated in relation to the safe procedure, and result feedback (**b**), which only stated whether the safe procedure was complied with.

### 2.3.2. Audit Consequence: Reward versus Punishment

In both audit consequence conditions, the participants received a message after an audit had been conducted. Positive audit feedback appeared if the participant had complied with the rule, and negative audit feedback appeared if the software had detected a rule violation. The two conditions (reward and punishment) differed in terms of the consequences of positive or negative feedback.

In the punishment condition, the participants lost all of the money they had earned in the trial in which the violation was detected. If they complied with the rule, they received a thank-you message.

In the reward condition, there was no financial punishment after negative audit feedback. If the auditing software implemented in WatrSim detected the application of the safety-related rule, the participants received a financial reward.

The financial punishment and reward were calculated such that the advantage or disadvantage of violating or complying with the rule was balanced out (Table 1). Considering the consequences of a positive or negative audit in the respective condition (punishment or reward), there was no financial difference within the condition if participants always violated or always complied with the rule.

The punishment was calculated based on 9 audits in 36 relevant stages of the start-up procedure. The maximum salary that could be achieved with rule compliance was EUR 0.80, for a total of EUR 28.80. If the rule were consistently violated, a maximum of EUR 1.00 could be earned, resulting in 36.00€. The punishment of losing the salary of the respective stage (depending on the participant's performance, with a maximum of EUR 1.00) per detected violation resulted in (maximum) EUR 9.00 (if the rule was consistently violated and the participant performed the procedure correctly). This led to a salary of approx. EUR 27.00, meaning that the advantage of violating the rule disappeared.

**Table 1.** Calculation of salary if participants always violate or always comply with the rule in the punishment and reward conditions.

| | Before Intervention 1st Quarter | After Intervention 2nd–4th Quarter | Financial Difference of Compliance vs. Violation | Financial Difference between Punishment and Reward | Potential Maximum Salary |
|---|---|---|---|---|---|
| Compliance | 12 × 1€ = 12 | 36 × 0.8€ = 28.80€ | | **Punishment**: no punishment | 40.80€ |
| | | | 7.20€ | **Reward**: 9 × 0.8€ =7.20 | 48.00€ |
| Violation | 12 × 1€ = 12 | 36 × 1.0€ = 36.00€ | | **Punishment**: 9 × no salary | approx. 40€ |
| | | | | **Reward**: no reward | 48.00€ |

The reward took into account that rule compliance had a financial disadvantage of EUR 7.20 (EUR 28.80 vs. EUR 36.00). This amount was distributed among the 9 audits, with a EUR 0.80 bonus payment per positive audit feedback. If a participant consistently complied with the rule, a financial disadvantage of EUR 7.20 was therefore compensated.

### 2.3.3. Feedback Communication: Loss versus Gain

The WaTrSim interface was designed to frame information on salary and productivity. In the gain-framing condition, the actual amount of money earned was depicted.

In contrast, in the loss-framing condition, the interface looked identical but displayed the difference from the maximum possible salary (EUR 1.00). The same salary of EUR 0.40 in the gain framing was depicted in the loss framing but presented as EUR −0.60. With regard to productivity, both numerical and graphical depictions were provided, both showing the desired and actual values. The desired value (thin "target line") represents the maximum output that can be achieved by conducting the safe procedure and without violating the safety rule (Figure 1, bottom right).

### 2.3.4. Work-Safety Tension

The perceived work-safety tension was measured with a four-item scale ([45], Table 2). This was originally measured with a 7-item scale with two facets. We included only the facet of "barriers to safety compliance", which describes the core characteristics of the work-safety tension with regard to the effects of goal conflicts on safety-related behavior.

### 2.4. Dependent Variables

The number of violations represents the sum of unsafe procedures counted in simulated quarters 2–4. The system detected whether the unsafe or safe procedure had been conducted and delivered an output dataset, in which 0 represents compliance (safe procedure) and 1 represents violation (unsafe procedure).

We also measured the output in liters of purified water in terms of actual productivity. The unsafe procedure enabled an output of 199 L of the separated solvent and water mixture within the two minutes available for production in one simulated week. The application of the extended and safe procedure led to a slower start-up and lower production, which, due to the three additional steps, was limited to 147 L within the two minutes available.

The production outcomes indicate the extent of rule compliance, as there is a wide range of possible production outcomes to be achieved by conducting the unsafe or safe procedure. As described in the process feedback section, even if the rule were complied with, the respective procedure could still contain some actions from which the operator (participant) might deviate. The range of achievable production outcomes depended on the chosen rule-related strategy, which we identified in previous studies [50,51].

## 2.5. Demographic and Person-Related Variables

For our recent investigation, besides standard demographic variables, we also assessed a set of control variables, which we had identified as relevant in the context of safety-related rule violation and for learning and recalling procedures in previous studies [22,23,32].

As demographic variables, we recorded age, sex, Abitur grade (German university entrance-level examinations), course of studies, and number of completed semesters. Table 2 describes the questionnaires that we used to measure the control and dependent variables.

**Table 2.** Descriptions of questionnaires used to collect information regarding control and dependent variables.

| Variable | Description | Measure | Example Items (Translated, Originally in German) |
|---|---|---|---|
| General mental ability (0–50) | General mental ability speed test | Wonderlic-test [52] 50 items, 12-min limit | "What is the next number in this series?" 1–0.5–0.25–0.125–? |
| Prior knowledge (0–7) | Relevant knowledge about wastewater treatment plants and technical specifications | Self-generated [22,23] 7 items, 1 P. for each correct answer | "What does homogenization mean?" |
| Knowledge of safe procedure (0–7) | Procedure description of each of the 13 steps with fill-in-the-blanks | Knowledge test, 7 blanks to fill in, 1 P. for each correct fill | 4. ▭ |
| Knowledge of unsafe procedure (0–5) | Procedure description of each of the 11 steps with fill-in-the-blanks | Knowledge test, 5 blanks to fill in, 1 P. for each correct fill | 9. ▭ |
| Feedback acceptance (1–5) | Behavioral effectiveness of the feedback | Self-generated, 3 items | "I have tried to implement the feedback we received." |
| Work-safety tension (1–5) | Sensing tension between productivity and safety | Work-safety tension [45], 4 items | "Sometimes it is necessary to deviate from safety regulations because of productivity." |
| Everyday dilemma (1–4) | Rule violation in everyday situations | [50], 10 items | "I'd rather risk being caught speeding than be late for an important appointment." |
| Feedback perception (1–5) | Facets of feedback (evaluation, result, medium) | Adapted from Huang et al. [53], 19 items | "I have received sufficient feedback on my activities." |

The questionnaires that address cognitive aspects are a general mental ability assessment and prior knowledge test as well as knowledge tests on the procedures taught to the participants in the study, which are explained in the following section (safe and unsafe procedures). With regard to the behavioral level, the questionnaire on everyday dilemmas asks about the general handling of safety-related goal conflicts in different everyday situations. The effectiveness of the received feedback was evaluated by the feedback acceptance questionnaire. In addition, we recorded different aspects of perception. The feedback perception describes the assessment of the feedback results (positive or negative) as well as the preferred medium through which feedback is conveyed.

## 2.6. Experimental Procedure

An overview of the experimental procedure is provided in Table 3. Upon arrival, participants were greeted by the experimenter and then underwent a time-restricted paper-and-pencil test to measure general mental abilities. The other variables were gathered using an online-based questionnaire. Next, the experimenters introduced the core functions and the interface of WaTrSim and demonstrated the unsafe procedure (11-step procedure). As support for practicing the start-up procedure, the participants received a handbook containing a description of the production goal (produce as much as possible and as fast as possible) and a list of the steps of the procedure. Participants were guided step-by-step through the application of the procedure. Subsequently, they practiced the procedure four times. Next, the handbook was taken away, meaning that participants had to demonstrate their skill in executing the procedure without help. After this, the first simulated quarter began with the first 12 simulated production weeks.

**Table 3.** Overview of the sequence of actions during the experiment.

| | |
|---|---|
| Reception | 5 |
| Measurement of control variables I<br>Demographic variables, prior knowledge, and general mental abilities | 30 |
| Introduction and training of WaTrSim<br>Core functions and standard (unsafe) procedure | 50 |
| Performance and knowledge test<br>Standard (unsafe) procedure | 15 |
| System operation<br>1st quarter | 25 |
| Break and intervention<br>Deflagration and management directive | 10 |
| Training WaTrSim<br>Extended (safe) procedure | 25 |
| Performance and knowledge test<br>Extended (safe) procedure | 15 |
| Announcement of audits | 5 |
| System operation<br>2nd–4th quarter | 75 |
| Measurement of control variables II<br>Manipulation check, presence, feedback acceptance and perception, self-interest, cautiousness, regulatory focus, work-safety tension | 30 |
| Debriefing and farewell | 5 |
| | $\sum$ = approx. 290 min |

During a break after the first quarter, the participants were asked to leave the room. The experimenter then informed participants of a management directive stating that a deflagration in the (fictional) company had destroyed a part of the plant. To avoid such events in the future, the prior procedure was now deemed to be unsafe, and the new, safe procedure was declared mandatory. The consequence of a violation (punishment condition) or compliance (reward condition) was described depending on the audit consequence condition (punishment or reward). Next, the investigator explained and demonstrated the new, extended procedure, before the participants conducted the safe procedure twice with the help of a handbook (which described the procedure step-by-step). The handbook was then taken away, and the participants had to demonstrate their skill in executing the safe and mandatory procedure without any help.

During the reception phase, the experimenter asked the participants to introduce themselves by stating their name, course of study, and how they became aware of the study. This was used to check their language abilities. Although it was communicated during study recruitment that fluent German language skills were necessary to participate in the study, some potential participants showed substantial language deficiencies, which would have made it difficult for them to understand the training content. If participants were unable to respond properly or had to use a dictionary, they were politely asked to withdraw from the study.

## 3. Results

Data were collected from 185 participants. Of these, 19 were excluded from the analysis because they did not meet the inclusion criterion of having sufficient skills to conduct the

trained procedure before starting the production phase, as defined by a production outcome of above zero in the fourth training week. Thus, N = 166 participants were included in the analysis (age $M$ = 22.70 years, $SD$ = 3.17, 54 female). In order to determine whether the collected control variables could be included as relevant covariates in the analysis, the preconditions were first checked. Due to their correlation with the DV, the variables Work-Safety Tension, Everyday Dilemma, Positive and Negative Feedback Perception, and Feedback Display are considered. None of these variables meet the requirements of normally distributed residuals according to the Kruskal–Wallace test. However, due to the central limit theorem and the size of the sample, we assume that the violation of the condition does not have a significant impact on the results.

As displayed in Table 4, the number of violations was correlated with the production outcome, which generally supports the experimental design to induce a goal conflict between safety and productivity that can only be resolved by choosing the safe or productive approach. Apart from this, a high number of violations was only associated with perceived work-safety tension and a general tendency to violate rules in everyday life, but this tendency did not lead to a higher production outcome. Participants with a higher tendency to violate rules in everyday life also perceived more work-safety tension. The production output was affected by knowledge, which, in turn, was unrelated to the number of violations.

Regarding the person-related variables, cautiousness (higher values indicate less cautiousness) correlates with chronic regulatory focus and everyday dilemma, supporting the criterion-related validity of the chosen measures. Similarly, the validity of self-interest was demonstrated by its correlations with work-safety tension and everyday dilemma.

### 3.1. Hypothesis Testing

As recent studies have suggested that F-tests are robust to non-normality if the compared group sizes are equal [54], we conducted an ANOVA using SPSS 24, although the dependent variable of number of violations was not normally distributed ($p < .01$). Regarding the independent variables of feedback mode (H1: more violations with result feedback than with process feedback), feedback consequence (H2: more violations after punishments than after rewards) and framing (H3: more violations in loss than in gain framing), ANOVA revealed a significant main effect for the feedback mode ($F_{(1,158)}$ = 4.14, $p < .05$, $eta^2$ = .03) but no main effect for framing ($F_{(1,158)}$ = 1.07, $p$ = .30, $eta^2 < .01$) or for audit consequence ($F_{(1,158)}$ = 0.65, $p$ = .42, $eta^2 < .01$).

The correlation of work-safety tension with the number of violations (H4: the higher the work-safety tension, the greater the number violations that occur) was significant and positive ($r$ = .18, $p < .05$), thus confirming Hypothesis 4. If the participants perceived that the organization valued productivity more than safety, even though the opposite was officially communicated, their own motivation to follow the safety rule was undermined, and the tendency to violate this rule increased.

Regarding Hypothesis 1, contrary to our assumption, participants in the process feedback condition violated the rule 1.5 times more often ($M$ = 7.47, $SD$ = 10.38) than participants in the result feedback condition ($M$ = 4.72, $SD$ = 7.41). Therefore, Hypotheses 1–3 must be rejected.

We can conclude that, all other influences being equal, participants in the gain and loss conditions violated the rule to the same extent. The same also applies to the audit consequence. Irrespective of whether participants were rewarded for complying with the rule or punished for violating it, the degree of rule violation shown in these conditions did not differ. The only main effect was related to the feedback mode: the repeated process feedback, which showed the actual conducted procedure rated in relation to the safe procedure, resulted in a higher number of rule violations than if only the result feedback was displayed, which merely stated whether the safe or unsafe procedure had been conducted.

Table 4. Bivariate correlation matrix between dependent and control variables.

| Variable | M | SD | α | 1 | 2 | 3 | 4 | 5 | 6 | 7 | 8 | 9 | 10 | 12 | 16 | 17 | 18 | 19 | 20 | 21 |
|---|---|---|---|---|---|---|---|---|---|---|---|---|---|---|---|---|---|---|---|---|
| Violation (1) | 6.10 | 9.10 | | | | | | | | | | | | | | | | | | |
| Output (2) | 137.72 | 31.67 | | .46 ** | | | | | | | | | | | | | | | | |
| Sex (3) | 1.67 | 0.47 | | .10 | .28 ** | | | | | | | | | | | | | | | |
| Age (4) | 22.70 | 3.17 | | −.15 | −.15 * | .16 | | | | | | | | | | | | | | |
| Abitur (5) | 2.52 | 2.29 | | .01 | −.02 | −.13 | .05 | | | | | | | | | | | | | |
| Semester (6) | 4.59 | 3.50 | | −.12 | −.09 | −.07 | .51 ** | .17 * | | | | | | | | | | | | |
| General mental abilities (7) | 25.91 | 5.93 | | .08 | .39 ** | .10 | −.15 * | −.00 | .03 | | | | | | | | | | | |
| Prior knowledge (8) | 5.83 | 1.21 | | .07 | .13 | .17 * | −.06 | −.00 | .10 | .30 ** | | | | | | | | | | |
| Knowledge test safe procedure (9) | 6.84 | 0.43 | | −.11 | .16 * | .00 | .04 | −.02 | .11 | .32 ** | .14 | | | | | | | | | |
| Knowledge Test unsafe procedure (10) | 4.72 | 0.57 | | −.13 | .14 | .08 | .10 | .03 | .13 | .16 * | .08 | .30 ** | | | | | | | | |
| Feedback acceptance *** (12) | 3.84 | 0.94 | .80 | −.20 | −.11 | −.25 * | −.17 | −.11 | .13 | .01 | .05 | −.06 | −.16 | | | | | | | |
| Work-safety tension (16) | 2.71 | 0.88 | .66 | .18 * | .19 * | .05 | −.13 | −.08 | −.03 | .05 | −.11 | −.06 | .01 | −.16 | | | | | | |
| Everyday dilemma (17) | 1.82 | 0.39 | .56 | .19 * | .05 | .04 | −.04 | −.11 | .13 | −.00 | .03 | .06 | −.05 | −.07 | .41 ** | | | | | |
| Feedback positive (18) | 3.46 | 1.08 | | −.34 ** | −.17 * | −.01 | .11 | .08 | .14 | −.08 | −.05 | −.08 | −.11 | .46 ** | −.19 * | −.19 * | | | | |
| Feedback negative (19) | 2.26 | 1.08 | | .33 ** | .16 * | .17 * | −.16 * | −.09 | −.18 * | −.02 | −.06 | −.02 | −.04 | −.49 ** | .30 ** | .22 ** | −.48 ** | | | |
| Feedback voice (20) | 2.16 | 1.21 | | .04 | −.15 | .10 | .04 | −.07 | −.15 * | −.29 ** | .02 | −.06 | .03 | −.18 | .04 | −.09 | −.07 | .01 | | |
| Feedback display (21) | 3.46 | 1.13 | | −.20 ** | −.03 | .01 | .02 | .06 | .11 | .03 | .01 | .04 | .01 | .36 ** | −.12 | −.20 * | .20 ** | −.30 ** | −.20 ** | |
| Feedback printout (22) | 3.22 | 1.17 | | .06 | −.00 | .02 | .03 | .06 | −.06 | −.05 | −.01 | .02 | .00 | −.09 | −.06 | .01 | .05 | −.10 | .18 * | −.01 |

Note: Sex: 1 = female, 2 = male; * $p < .05$, ** $p < .01$, N = 166 except *** N = 78.

### 3.2. Post-Hoc Analysis

Furthermore, we investigated whether the main effect interacted with any of the other factors. Of all possible and sensible combinations, the analysis revealed a significant interaction between feedback mode and framing ($F_{(1,158)} = 7.26$, $p < .01$, $eta^2 = .04$). Whereas there was no greater difference between the participants who received result or process feedback under the loss condition (result: $M = 5.85$, $SD = 8.17$; process: $M = 4.97$, $SD = 7.93$), the participants in the gain condition differed remarkably (Figure 3) in the number of rule violations depending on whether they received result feedback ($M = 3.61$, $SD = 6.48$) or process feedback ($M = 10.10$, $SD = 11.97$). In other words, the repeated feedback regarding performance standards, realized as process feedback, led to a higher tendency to violate safety-related rules only if the monetary outcome was framed as a gain. If the monetary outcome was displayed as a loss in relation to the maximum possible monetary outcome, the process feedback had no effect on violations. Thus, workers perceived the information of the process feedback as guidance to improve their strategy of conducting the start-up in order to achieve greater output and salary.

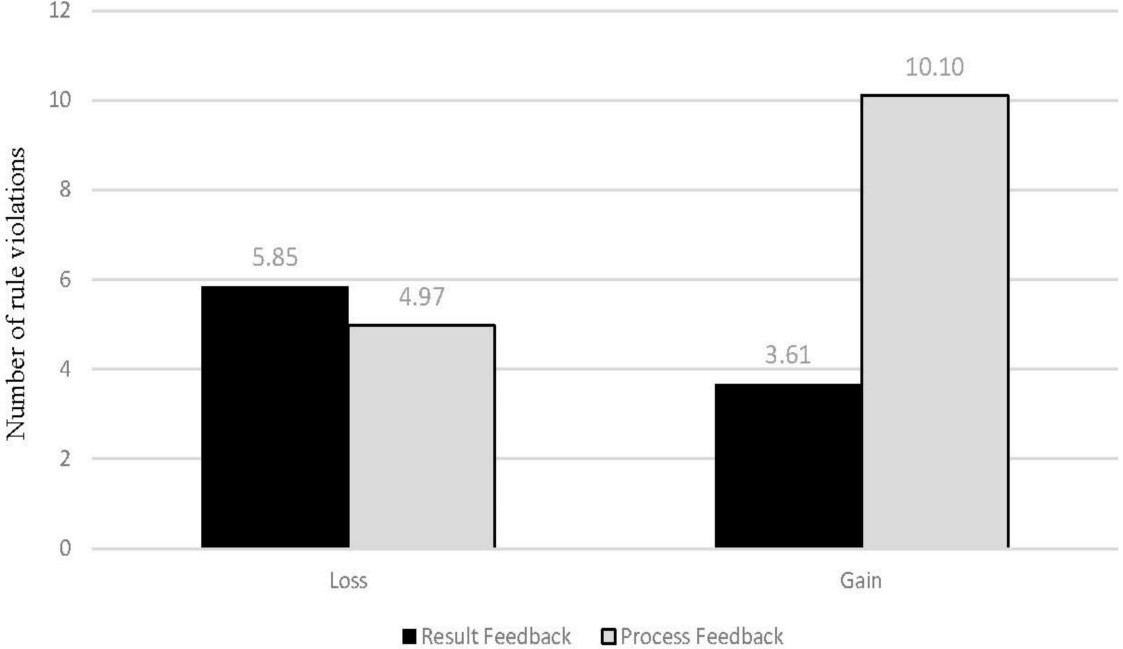

**Figure 3.** Interaction effect of the framing (loss and gain) and feedback mode (result and process) factors.

The results of the meta-analysis by Pichler et al. [31] led us to analyze the reaction to the audits (as result and process feedback) in more detail. In previous studies, we investigated the bomb crater effect, which describes a rule violation that is committed directly after a conducted audit [44] and is termed a negative feedback reaction for the purpose of the present study. As a first step, we tested for a correlation between feedback acceptance and negative feedback reaction in order to check whether the bomb crater effect was related to attitudes towards feedback. The questionnaire for feedback acceptance was integrated into the study later on, which is why the sample is smaller. We found a significant negative correlation (N = 78, $r = -.27$, $p < .05$) in which the higher the feedback acceptance, the lower the probability of negative feedback reactions in terms of rule violations committed directly after a conducted audit. Given Pichler et al.'s [31] finding that feedback appraisal reactions are influenced by the knowledge of performance standards, we expected to find an even stronger effect of feedback acceptance on negative audit reactions if the participants received process feedback compared with result feedback. In fact, when we calculated the correlation only with the participants in the process feedback condition, no correlation emerged (N = 48, $r = -.03$, $p = .82$), whereas we found a medium correlation for

participants in the result feedback condition (N = 30, $r = -.52$, $p < .01$). In other words, if participants have no knowledge about the performance standards (result feedback), the feedback acceptance is important for the feedback reaction: if the acceptance is low, this might result in increased violations after audits. When process feedback is provided, which imparts knowledge about performance standards, the feedback acceptance is less relevant to the behavioral reaction to the feedback.

The medium correlation between feedback acceptance and the demand for displayed feedback (Table 4 $r = .36$, $p < .01$) supports the importance of a tailored feedback design. Furthermore, participants who wished for displayed feedback also violated rules less frequently when it was delivered (Table 4, $r = .20$, $p < .01$).

When examining the perception of feedback, a further interesting aspect emerged with respect to the reception of positive or negative audit feedback: participants who perceived the feedback as positive had a higher feedback acceptance ($r = .46$, $p < .01$) and reported less work-safety tension. Participants who perceived the feedback as more negative had a lower feedback acceptance ($r = .46$, $p < .01$).

## 4. Discussion

The present study aimed to contribute to the debate about the effectiveness and the effects of behavior-based safety measures on safety performance with empirical results and to derive suggestions for designing a safety-oriented work environment and effective safety communication. It is important to highlight that all of the results must be interpreted in the light of goal conflicts implemented in the simulation. Within this framework, an important precondition for an effective behavior-based safety program is violated; namely, the commitment of the management to safety, but this also depicts realistic occupational circumstances [55–57]. Rather than achieving our original intention of clarifying existing issues, the key finding of the present study injects a further aspect into the debate, as we found interacting effects of safety measures. First, we discuss how and why process feedback increases the violation of safety-related rules if the goals are gain-framed. Second, we discuss the role played by attitudes towards feedback and knowledge in a negative feedback reaction, referring to a violation committed directly after an audit, also known as the bomb crater effect.

One of four hypotheses were supported in the present investigation, as the opposite of the assumed effects emerged. The frequently confirmed effect of gain-framing transformed from a perceived risk aversion into an opportunity if the feedback provided information about the procedure. We suggest two explanations for this, which support one another. First, the information provided by the feedback may have increased self-efficacy [58]. Krueger and Dickson [59] showed that perceived self-efficacy increases risk-taking behavior, which is supported by our results. The integrated model of behavioral prediction applied to violations (deleted for review) includes control beliefs as a factor of self-efficacy, influencing risky decision-making processes.

The second explanation lies in the cognitive cost–benefit tradeoff that emerges in decision-making processes. Using fMRI, Gonzalez et al. [60] demonstrated that the cognitive effort required to choose sure gains was lower than that required for risky gains, whereas risky and sure losses did not differ. In our study, gains and losses became sure or risky based on the amount of information provided, implemented as process (sure) and result (risky) feedback, to support the decision-making process. The results of Gonzalez et al. [60] perfectly correspond to our results concerning violations in the gain or loss framing under consideration of the given feedback (Figure 3). The sure gain (process feedback under the gain-framing condition) had the lowest cognitive cost and thus turned a risky decision into an opportunity: these low cognitive costs may be interpreted as self-efficacy, leading to the risky decision (interpreted as an opportunity).

These findings emphasize that any safety measure can be a slippery slope if there are goal conflicts between safety and productivity. So far, gain framing and process feedback have been found to be good measures for reducing risk-taking behavior. However, under

the pressure of goal conflicts, they interact counterproductively, although we did find some evidence that process feedback affected the tendency to violate rules after a conducted audit. If the feedback itself is not well accepted, the process feedback can be used to compensate for the negative effect of low feedback acceptance on risk-taking behavior. Although feedback acceptance was not higher for those with process feedback than with result feedback, the effect of low acceptance did not materialize when process feedback was provided. Our findings regarding the measured person-related variables give rise to the question of when feedback might be accepted and when feedback might lead to reactant behavior. A preliminary conclusion might be that the mere result of the feedback, i.e., whether it is positive or negative, could make the difference.

### 4.1. Strengths and Limitations

In order to identify interacting safety measures, it was important to design a study that considers several factors influencing safety-related behavior and, at the same time, ensures internal validity. This notable strength of the present study was realized by using a simulated plant environment. A simulation-based study is the only reasonable approach to investigating the effects of manipulated feedback types on safety-related work behavior. In addition, the experiment allows us to control for influencing factors, which is not possible in this form in a company. However, these advantages are accompanied by one of the studies' weaknesses; namely, the sample size in each condition. Achieving a minimum number of required participants consumes a high degree of resources, given the duration of the study. Recognizing that the scope of the experiment makes it difficult to enroll enough participants, it is important to note that the relatively small number of participants might have influenced the results and, hence, the conclusions of the study. Based on the power calculation, the conclusions are to be considered as a very conservative interpretation. However, since the rejected hypotheses did not differ very clearly, we consider the type 2 error to be less likely. Furthermore, only engineering students were eligible to participate in the study, which reduces the number of eligible participants and therefore complicates the acquisition of the findings but ensures their external validity. Through these tradeoffs, we reached a good balance between the demands for a reliable study and valid findings. With our findings for Hypothesis 4, regarding the impact of perceived work-safety tension, we were also able to show that the behavioral reaction in the experimental laboratory setting is comparable to the findings of field studies. This also supports the study's validity and allows us to draw conclusions and derive practical implications from our findings. In addition to this, we note that the deliberately induced goal conflict is a major driver of the effects shown. Further studies have already been conducted to investigate the influence of feedback, consequences, and framing with and without goal conflict. The results of these studies are currently being prepared for publication.

From a more general perspective, this study addresses a need in safety management research by spanning the gap between safety management and empirical research [61]. If we compare the results of this experimental study with the results of field studies, some parallels can be seen. Field studies among mineworkers show a link between dissatisfaction and accidents at work and call on safety managers to address the causes of risky work behavior [62], as we have outlined in our study. Furthermore, it was shown that the safety climate mediates trust in the organization and thus influences the injury rate [63]. It is clear from all perspectives that management's attitude towards safety has a significant influence on the safety behavior of workers. In particular, restrictions on safety management for economic reasons have a disastrous influence on safety behavior [64].

### 4.2. Implications for Further Research

With regard to the bomb crater effect, we were able to contribute some findings to the question of its function. The debate concerns loss repair (violating behavior after a conducted audit to compensate for losses suffered from the audit result) or misperception of chance (underestimation of the probability of consecutively conducted audits [65]). In

a previous study, we found evidence that the bomb crater effect has the function of loss repair [44]. If feedback acceptance is also important for the reaction to audit results, the expression of sensed fairness and legitimacy might be a further function of the bomb crater effect that has not yet been discussed. Furthermore, the link between self-interest and feedback acceptance and perception is worthy of further research.

Our study also highlighted the need for complex study designs in order to depict the complex conditions of a working environment. Studies that simplify investigations to single aspects turn a blind eye to interacting effects.

## 5. Conclusions

For the design of behavior-based safety programs, a central result of the present study is the repeated finding that a goal conflict between safety and productivity always culminates in the violation of safety-related rules. The first question with regard to the design of safety measures is where goal conflicts are initially created, and the answer is the top-level management of an organization. If front-line workers perceive work-safety tension, they tend to prefer the productive route, implicitly requested by the organization but requiring unsafe behavior. Our results also show that the feedback of audit results should contain information about the audited process. Otherwise, the reaction to this feedback depends on its acceptance and could lead to reactant behavior, meaning that violations are committed directly after the audit feedback is delivered. To be on the safe side, process feedback should be provided, which involves high acceptance. To enhance acceptance, the feedback design should be based on workers' wishes with regard to how feedback is presented. As our results show, if participants wished for displayed feedback, their acceptance was higher, and their violation rate was reduced when such feedback was provided. The results also indicate that behavioral safety cannot be achieved through mere incentives or compliance with regulations. All measures taken should take place within a coordinated concept and focus on concrete safety-relevant aspects of the work environment.

The most important message for practitioners lies in the insight that safety measures interact, and this interaction can turn their individual positive effects into an overall negative one. Although we were able to demonstrate that process feedback has positive effects in terms of preventing unsafe behavior after a delivered audit result, it can also have harmful effects if it is combined with gain-framed production goals. We do not suggest a loss-framed environment with mere result feedback, as the loss framing of production goals has its own pitfalls. Rather, the negative effects refer to the underlying goal conflict, which should be eliminated first.

The second most important insight is the resulting follow-up question: what happens without goal conflict? This is the question we will be looking at in future studies. Our preliminary assumption is that the organization's goal conflicts have a direct impact on workers, making incidents and accidents based on safety breaches likely. Part of the responsibility for incidents and accidents remains with the employer as long as the employer is responsible for goal conflicts. This includes investors and the design of work processes.

**Author Contributions:** Conceptualization, A.K. and S.B.; methodology, A.K. and S.B.; software, A.K. and S.B.; validation, A.K. and S.B.; formal analysis, S.B.; investigation, S.B.; resources, A.K.; data curation, S.B.; writing—Original draft preparation, S.B.; writing—Review and editing, A.K. and S.B.; visualization, S.B.; supervision, A.K.; project administration, S.B.; funding acquisition, A.K. All authors have read and agreed to the published version of the manuscript.

**Funding:** This work was supported by the German Research Foundation under Grant KL2207/2-3.

**Institutional Review Board Statement:** The study was conducted according to the guidelines of the Declaration of Helsinki, and approved by the Institutional Ethics Committee of the Faculty of Psychology at Ruhr University Bochum (protocol No. 189, 18.02.2015).

**Informed Consent Statement:** Informed consent was obtained from all subjects involved in the study.

**Data Availability Statement:** The data presented in this study are available on request from the corresponding author. The data are not publicly available due to participants data protection policy.

**Acknowledgments:** We would like to thank the Ruhr-University Bochum for its support and cooperation, as well as the university alliance partners TU Dortmund and University Duisburg-Essen for providing facilities to acquire and conduct the study. In particular, we would like to thank the assistants Merle Lau, Lina Kluy, Felix Miesen, Pia Schempp, Leonie Kloep, Maike Puhe, and Lena Iffland, who were conscientiously and persistently involved in data collection and data maintenance over the years. For the competent and patient extension of our WaTrSim software we thank Kathrin Bischof. We are especially grateful for the appreciative and constructive support of the German Research Foundation (DFG).

**Conflicts of Interest:** The authors declare no conflict of interest.

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
