# Peer review of "When the Tension Is Rising: A Simulation-Based Study on the Effects of Safety Incentive Programs and Behavior-Based Safety Management"

_safety, 2020_

Round 1

Reviewer 1 Report

Authors improved the quality and style of the paper, addressing the major part of reviewers' comments.

The manuscript is well written, technical content is adequate, the methodology and results are clearly described and commented

Only few minor observations:

In Introduction authors correctly recall the role of safety rule violation as root cause of notable accidents, quoting a number of papers. The paper sub ref. 1 seems more indicated as reference to Seveso accident that can be quoted explicitly as well, rather than for Bhopal. The last one was more recently re-investigated with more pertinent human and managerial details e.g. in the suggested http://dx.doi.org/10.1016/j.psep.2015.06.009  , while the paper sub ref. 2 is not useful, as it deals with inherent safety approach, that is outside of the topics of the manuscript.

Looking at the list of refs, there are some missing information, compared to the requested format of the journal, e.g. ref. 9 should include the title: "Occupational Health and Safety Indicators and Under-Reporting: Case Studies in Chinese Shipping". Please check all refs. and complete.

Author Response

Note: In Introduction authors correctly recall the role of safety rule violation as root cause of notable accidents, quoting a number of papers. The paper sub ref. 1 seems more indicated as reference to Seveso accident that can be quoted explicitly as well, rather than for Bhopal. The last one was more recently re-investigated with more pertinent human and managerial details e.g. in the suggested http://dx.doi.org/10.1016/j.psep.2015.06.009  , while the paper sub ref. 2 is not useful, as it deals with inherent safety approach, that is outside of the topics of the manuscript.

Resp: Thank you very much for your comprehensive review. We have taken your suggestions into account and have included the paper you recommended. Since the information regarding the second source you mentioned is also shown there, we have decided to delete the second source without replacement.

Note: Looking at the list of refs, there are some missing information, compared to the requested format of the journal, e.g. ref. 9 should include the title: "Occupational Health and Safety Indicators and Under-Reporting: Case Studies in Chinese Shipping". Please check all refs. and complete.

Resp.: We have intensively reviewed the literature list and implemented your and other comments in this regard.

Reviewer 2 Report

The authors have done a nice job responding to the prior reviews.  I think this is an interesting study that should be of interest to researchers and practitioners seeking to develop programs to reduce the negative impact of work-safety tension on safety-related outcomes. 

Author Response

Note: The authors have done a nice job responding to the prior reviews. I think this is an interesting study that should be of interest to researchers and practitioners seeking to develop programs to reduce the negative impact of work-safety tension on safety-related outcomes. 

Resp.: Many thanks for your appreciative feedback.

Reviewer 3 Report

This is a very well written manuscript! Well structured, fairly up to date references (many older), appropriate keywords (no overlap with terms in the title :). The following are suggestions for improving the manuscript:

Abstract: Prioritize text (due to word limit) that reveals what is new in this study and the practical impact on industry. The abstract needs to be more appealing as to how this manuscript is furthering knowledge and practice for the age -old challenge with work-safety tension. This will help increase readership and citations of the article. E.g. in the BBSM paradigm - is your result of "workers' wishes' as to how feedback is presented" (line 567) a key result for future practice/theory?

Introduction: Discuss/present how safety (health and wellbeing) should be, according to laws and regulations, an integrated part of business and professions, yet often ends up in a safe and separate, competing silo. 

The four hypotheses are much more clearly and concisely stated in the discussion (lines 412-420). Consider adding these short descriptions to the hypotheses section (1.3); e.g. (H1: more violations with result feedback than with process feedback).

Method: Provide additional reasons for reducing the sample size in spite of power calculations showing the need for a large sample size. Is it responsible science to continue a study despite low power? Likewise in the limitations - further discuss whether this may have contributed to the non-significant results.

Discussion and recommendations: Where do the authors feel the best return on research investment is in OBM (management, systems, behaviour) - if the three areas should be prioritized? Could focus be on leadership (safety) KPIs? Are worker's work-safety tensions reflective of leader's work-safety tensions? Owners? Executive Boards? Stockholders? 

References: Avoid use of capital lettering e.g. reference 3, 10, 12, 14, 28, 38, 59 etc.; Add title to Xue C et al 2019

Again, thank you for submitting a well-written manuscript - a pleasure to read. 

Author Response

Note Abstract: Prioritize text (due to word limit) that reveals what is new in this study and the practical impact on industry. The abstract needs to be more appealing as to how this manuscript is furthering knowledge and practice for the age -old challenge with work-safety tension. This will help increase readership and citations of the article. E.g. in the BBSM paradigm - is your result of "workers' wishes' as to how feedback is presented" (line 567) a key result for future practice/theory?

Resp. Thank you for your detailed feedback. We highlighted the important points you mentioned and reduced some other less important information. In the text (see attachment) we only highlighted the newly inserted aspects. All reduced text is already deleted.

Note Introduction: Discuss/present how safety (health and wellbeing) should be, according to laws and regulations, an integrated part of business and professions, yet often ends up in a safe and separate, competing silo.

Resp.: This is a very inspiring suggestion. The question of the cause of unsafe behavior also raises the question of why there must be laws and regulations for safe behavior at all. As important and relevant as we find this question, this paper does not seem to have the adequate place for it. We fully agree that the question is of great importance and that the setting under investigation provides the framework for this discussion. However, an appropriate consideration of this question seems to us to go beyond the scope of this paper. We consider to dedicate a separate chapter to this question.

Note: The four hypotheses are much more clearly and concisely stated in the discussion (lines 412-420). Consider adding these short descriptions to the hypotheses section (1.3); e.g. (H1: more violations with result feedback than with process feedback).

Resp.: Thank you for drawing our attention once again to our hypothesis formulation. As a result, we have noticed that the first hypothesis is indeed somewhat bulky. For the sake of simplicity we have simplified and standardized the first hypothesis according to the other hypotheses. We refrain from adding a supplementary short form, since the other hypotheses are already understandable and well abbreviated. We hope to have met the aim of this note as well as possible.

Note Method: Provide additional reasons for reducing the sample size in spite of power calculations showing the need for a large sample size. Is it responsible science to continue a study despite low power? Likewise in the limitations - further discuss whether this may have contributed to the non-significant results.

Resp.: We share the view that the cause and effect of the selected sample size should be considered with great care. The reasons described in the methods section were decisive for the decision, so we will refrain from adding further arguments in retrospect. However, we have added further remarks to the discussion of the limitations in order to convey to the readers the importance and consideration of the test power.

Note Discussion and recommendations: Where do the authors feel the best return on research investment is in OBM (management, systems, behaviour) - if the three areas should be prioritized? Could focus be on leadership (safety) KPIs? Are worker's work-safety tensions reflective of leader's work-safety tensions? Owners? Executive Boards? Stockholders?

Resp.: Many thanks for this very pointed suggestion. In a concluding paragraph we have taken up the idea of RORI (return on research investment, I like this term very much) and added a corresponding description.

Note References: Avoid use of capital lettering e.g. reference 3, 10, 12, 14, 28, 38, 59 etc.; Add title to Xue C et al 2019

Resp.: Thank you for this detailed review of the literature list. We needed some workarounds to convince the citation software to do what we want it to do.

Round 2

Reviewer 1 Report

Authors properly addressed comments and suggestions from reviewer, including improvement of the state-of-the-art, resulting in an appealing manuscript, weel worth publishing.

A very minor formal revision: in lines 29-31 and consequently 588-592 (Reference list), please correct the numbering of quoted papers according to the order of citation in the text. 

 Line 39 explicit the notation "BS" as "behavioural safety" (or add full text).

Author Response

Thank you for your appreciative and encouraging feedback. In the following I will gladly respond to your remarks.

Note:A very minor formal revision: in lines 29-31 and consequently 588-592 (Reference list), please correct the numbering of quoted papers according to the order of citation in the text.

Resp.: I was also able to remove the last Citavi bug with it. After final testing I did not notice any further irregularities. Thanks for the hint.

Note: Line 39 explicit the notation "BS" as "behavioural safety" (or add full text).

Resp.:Since the abbreviation is part of the title, I had not thought of writing it out. As a compromise solution, I added a note in brackets after the title "(author's note: BS = behavioral safety)". I highlighted the section in line 39 accordingly.

Reviewer 3 Report

The authors have suffficiently replied to all review-1 issues.

Please check and proofread the two (highlighted in yellow) newly revised/added texts in the abstract, which appear to be missing a word or making a term plural, i.e. 1) the connection between the first and second part of the sentence - should 'their' be 'the'?; 2) do you mean 'measures' (plural)

1) This allows us to directly observe the effects of goal conflicts and their countermeasures, we designed in this study.

2) We found acceptance of measure and their design as important for rule related behavior.

Author Response

Thank you for patiently and expertly reviewing the paper once again. In the following I will gladly respond to your remarks.

1) This allows us to directly observe the effects of goal conflicts and their countermeasures, we designed in this study.

Resp.: In this sentence I wanted to emphasize the relation of the countermeasures to the goal conflicts. With the revised formulation "This allows us to directly observe the effects of goal conflicts and of countermeasures, we designed in this study." I focused less on the reference, but rather on the fact that we can observe both the effects of goal conflicts and of countermeasures. In the tight corset of the abstract some words fall somewhat by the wayside.

2) We found acceptance of measure and their design as important for rule related behavior.

Resp.: Yes, it should say "measures". Please excuse the attentional error. I changed it accordingly.

Round 3

Reviewer 1 Report

Accept as is.

Author Response

Note: Accept as is.

Resp.: Thank you for your support. I am in complete agreement with you.

Reviewer 3 Report

The abstract in the 'online Abstract' submission box is not the same as the one in the submitted manuscript - please update. Both the updated sentences are missing.

Check as to whether the word 'the' should be added to the revised sentence:

"This allows us to directly observe the effects of goal conflicts and of the countermeasures, we designed in this study."

Author Response

Note: The abstract in the 'online Abstract' submission box is not the same as the one in the submitted manuscript - please update. Both the updated sentences are missing.

Resp.:

Thank you very much for the thorough review. I am not sure right now if I can change the abstract in the input mask for the resubmission. I won't see that until I submit this comment. If not, I will contact the editors about applying the change in the system.

Note: Check as to whether the word 'the' should be added to the revised sentence:

Resp.: Originally we had refrained from the non-essential definite articles due to the word limit. However, since we now have exactly two words remaining after the revisions, both "goal conflict" and "contermeasures" will get their own article.

This manuscript is a resubmission of an earlier submission. The following is a list of the peer review reports and author responses from that submission.

Round 1

Reviewer 1 Report

1. The authors use a clever and well-designed laboratory experiment to evaluate the effects of various safety incentive system design components and framing on safety-related behavior. Although experimental research can be critiqued due to its relative lack of ecological validity, it can be a valuable tool for evaluating causal mechanisms that would otherwise be difficult to ascertain in the field. Given this, I do believe that the article has the potential to make a valuable contribution to the extant literature. 

2. Please check for minor typos.  For example, p. 4(line 159), the sigma summation sign should be N (N=360). Also, for the Hypothesis 1, a colon is missing.  It should read "H1: xyz" rather than "H1 xyz". 

3. I realize that the authors deliberately chose to induce a goal conflict. However, setting up the study by telling participants they will be paid based on their productivity might have induced more work-safety tension than might be seen in real work settings, where organizational safety climate might influence the extent to which productivity is the sole driver of compensation.  Therefore, the authors might want to note this potential limitation to the generalizability in the Discussion.  In other words, would the same effects have been found if there was no goal conflict, or if safety and productivity were equally emphasized and rewarded?

4. I appreciate the authors specifically noting which analyses were post-hoc in nature. 

Reviewer 2 Report

Framing behavioral safety as "incentive" and "compliance" based is not consistent with the research and practice of the approach.  I am afraid this presentation is misleading and may be confusing to the reader.

The question of production tension is a good one and you found results.

The major issue I have with this manuscript is the scope of variables are confusing and cannot be clearly described or analyzed.  The study is framed as a three way design (2x2x2) which would require a three-way interaction that, I suspect, was indecipherable.  The myriad of person variables really confused the analysis and doesn't seem to add anything.  In the end, the large correlation and mean (SD) tables were hard to understand.  Also, the large amount of comparisons runs the risk of finding significance erroneously.  

Suggest a new manuscript revising the design down to the core IVs.  Perhaps other person variables can be in a different manuscript.

Reviewer 3 Report

The study is interesting and contains elemenys of novelty in the broad field of accident prevention and safety programs. The topic is well relevant to the Journal. Some modifications are suggested to increase the scientific appeal for the reader. Some parts of the paper are redundant, while some examples and clarifications should be added.

As for the contents, the paper meets the goals and the requirements for publication in "Safety".  Moreover, the paper contains a considerable amount of "experimental work" and the presentation is globally satisfactory.  In some cases, however, the results are presented in a somewhat acritical way.

1. Introduction covers main aspects, but would need some improvements. There are several notable accidents that include in the root causes under-reporting hazardous situations, unsafe behoaviour to maintain productivity and violations connected to economic pressure. At least following references should be added and commented in Introduction to enhance the appeal for the readers: 

Snow, N. 2010. BP marks 5-year anniversary of Texas city accident. Oil and Gas Journal 108, 35-37

Palazzi E., Currò F., Fabiano B., 2015, A critical approach to safety equipment and emergency time evaluation based on actual information from the Bhopal gas tragedy, Process Safety and  Environmental Protection 97, 37-48.

Kidama, N.E. Hussina, O. Hassan, A. Ahmad, A. Johari, M. Hurme. 2014. Accident prevention approach throughout process design life cycle. Process Safety and Environmental Protection 9 2, 412–422.

Moizés Martins Junior, Marcello Silva e Santos, Mario Cesar R. Vidal, Paulo Victor R. de Carvalho. 2012. Overcoming the blame game to learn from major accidents: A systemic analysis of an Anhydrous Ammonia leakage accident. Journal of Loss Prevention in the Process Industries 25, 33-39.

Point 1.3 of Introduction should be reduced in length and moved to Section 2. Material and methods.

2. Here some details are not needed. The technical and visual characteristics of WaTrSim can be amply sumarized, withot presenting a huge number of screeshots. 

3. Results are adequately presented, even if it is suggested contrasting better the results with similar field studies performed by other researchers. It would be interesting to extend this research beyond the academic exercise and try to provide some more implications of the results for translating them into real scale applications. Possibly implications can be different depending of the given industrial sector.  A summarizing table of main findings would be useful as well.

As already stated, the conflict productivity, economic pressure vs. safety can be better analyzed also with some practical examples from literature. Another item is under-reporting of near miss, violations etc. in case of award saefty program. 

After revision, a final check to English style is advisable.